# Validation of a Questionnaire Developed to Evaluate a Pediatric eHealth Website for Parents

**DOI:** 10.3390/ijerph17082671

**Published:** 2020-04-13

**Authors:** Bruno José Nievas Soriano, Sonia García Duarte, Ana María Fernández Alonso, Antonio Bonillo Perales, Tesifón Parrón Carreño

**Affiliations:** 1Nursing, Physiotherapy, and Medicine Department, University of Almería, Almería 04120, Spain; tesifonparron54@gmail.com; 2Obstetrics and Gynaecology Unit, Torrecárdenas Hospital, Almería 04009, Spain; sgarciaduarte@hotmail.com (S.G.D.); anafernandez.alonso@gmail.com (A.M.F.A.); 3Pediatrics Unit, Torrecárdenas Hospital, Almería 04009, Spain; abonillop@gmail.com

**Keywords:** eHealth, mHealth, pediatric, questionnaire, survey, evaluation

## Abstract

There is a need for health professionals to provide parents with not only evidence-based child health websites but also instruments to evaluate them. The main aim of this research was to develop a questionnaire for measuring users’ evaluation of the usability, utility, confidence, the well-child section, and the accessibility of a Spanish pediatric eHealth website for parents. We further sought to evaluate the content validity and psychometric reliability of the instrument. A content validation study by expert review was performed, and the questionnaire was pilot tested. Psychometric analyses were used to establish scales through exploratory and confirmatory factor analyses. Reliability studies were performed using Cronbach’s alpha and two split-half methods. The content validation of the questionnaire by experts was considered as excellent. The pilot web survey was completed by 516 participants. The exploratory factor analysis excluded 27 of the 41 qualitative initial items. The confirmatory factor analysis of the resultant 14-item questionnaire confirmed the five initial domains detected in the exploratory confirmatory analysis. The goodness of fit for the competing models was established through fit indices and confirmed the previously established domains. Adequate internal consistency was found for each of the subscales as well as the overall scale.

## 1. Introduction

The quality of parenting that children receive during their first years has a crucial influence on child development and health [1,2]. Parents need information about their children’s symptoms, and the Internet is a major resource [3]. Actually, it has changed the way that users search for information and take decisions about health [4]. The delivery of health services using information and communication technologies, especially the Internet, has been named eHealth [5], and among caregivers, the engagement and pursuit of health information online is higher than among the population in general [6]. Parents of sick children report using the Internet to find health information [7], and the Internet may serve as a convenient repository of health information [8]. Therefore, there is an urgent need for health professionals to provide parents with evidence-based child health websites [9].

On this subject, some of the proposals for the future of eHealth are to improve users’ health and eHealth literacy [10,11,12], to procure more health care providers’ implication in eHealth development [13], to improve usability of eHealth [14,15], and to investigate eHealth’s effectiveness [16]. We decided to develop and evaluate, from the point of view of the users, an open and free-access eHealth pediatric website for parents which contained 338 topics about pediatric symptoms, illnesses, and well-child care. The website was written by a pediatrician in Spanish and was certified by the Health on the Net Foundation in 2018. Unfortunately, we have not found validated questionnaires designed to evaluate pediatric websites for parents. The main aim of this research was to develop a questionnaire for accomplishing that task. We further sought to evaluate the content validity and psychometric reliability of the instrument.

## 2. Experimental Section

### 2.1. Study Design

A questionnaire was developed following the CHERRIES guidelines [17] (Appendix A), and reliability and validation analyses were performed [18]. The initial questionnaire’s main domains were accessibility, usability, trust and confidence, the well-child section, and utility of the website. The questionnaire contained 67 items; 26 were demographical items, and 41 were evaluation items (Appendix A). Responses for the items were generally four or five-point Likert scales. A website version of the questionnaire was developed using the Google Forms platform.

### 2.2. Content Validity, Sample Size, and Pilot Testing

A specific content validation for eHealth interventions was performed [16] based on expert review. Seven experts were recruited from different geographic locations of Spain. They were representative samples of the content domains of the questionnaire [19]. They were asked to evaluate both the website and the questionnaire, using a scale of 1 to 5 points to evaluate the core dimensions. They could also add open commentaries.

Following the recommendation of other authors [20] to calculate sample size, the Epi Info™ App, offered by Atlanta CDC (Epi Info™, Division of Health Informatics and Surveillance, Center for Surveillance, Epidemiology and Laboratory Services), Version 16 November 2018, available at https://www.cdc.gov/epiinfo/esp/es_index.html, was used with the following parameters: population size 2,432,167 (unique visitors of the website at that moment); 97% confidence interval, and a level of precision of estimate within 5% of either side of the true population proportion. These parameters indicated a required sample size of 471 participants. The authors decided to collect at least 500 responses, and eight weeks were needed to collect 516 valid questionnaires. This was in accordance with the classic rule established by Kline et al. [21] of using 2 to 20 subjects for each item of the questionnaire.

The questionnaire was made accessible through a link posted in twenty-five pages of the pediatric website. A specific page showed the purpose of the questionnaire and the main aims of the study. The feasibility, simplicity, and time required to answer the questionnaire were evaluated by five participants who did not take part in the research. Informed consent was shown on the initial page of the questionnaire. Only fully completed questionnaires were admitted. Personal data were not collected within the questionnaire or on the Google Forms platform. Google Forms generated a spreadsheet with the answers of the questionnaires. The data collected within that spreadsheet were used, without any manipulation or statistical correction, to perform the statistical analyses. No questionnaires needed to be discarded.

### 2.3. Construct Validity

The determination of the adequacy of the exploratory factor analysis (EFA) was performed through the analysis of Bartlett’s test and the Kaiser-Meyer-Olkin (KMO) measure. The KMO statistics range from 0 to 1, with values closer to 1 denoting greater adequacy of the factor analysis (KMO ≥ 0.6 low adequacy, KMO ≥ 0.7 medium adequacy, KMO ≥ 0.8 high adequacy, KMO ≥ 0.9 very high adequacy). If the result of Bartlett’s test is < 0.05, factorial analysis can be used. For the evaluation of construct validity, the 41 qualitative items of the questionnaire were submitted to the exploratory factor analysis.

Principal factor analysis with a Varimax rotation was used to explore the structure underlying the 41 qualitative items. The inclusion or exclusion of an item in a construct was determined iteratively by examining factor loadings and Cronbach’s alpha to identify redundant items or items that did not sufficiently measure the same underlying construct [22]. The inclusion or exclusion of an item in a construct was determined by factor loadings >0.6. Items with Pearson’s correlation coefficient <0.5 were discarded. Conceptually defined latent dimensions formed by the items were identified.

Confirmatory factor analysis (CFA) was performed by applying the maximum likelihood estimation method, available in AMOS software. The method was carried out to test the five-factor structures identified in the exploratory factor analysis. The goodness of fit for the competing models was evaluated through the most used fit indices [18]: the chi-square test, the goodness of fit index (GFI), the root mean square error of approximation (RMSEA), the normed fit index (NFI), the non-normed fit index (NNFI) or the Tucker–Lewis index (TLI), and the comparative fit index (CFI).

The reliability of the questionnaire was measured based on its internal consistency using Cronbach’s alpha [23]. As the questionnaire could not be retested with the same users, the split-half method was used.

### 2.4. Statistical Analyses and Review Board Approval

Statistical analyses were performed using SPSS version 26 (IBM Inc., Armonk, NY, USA). The statistical software AMOS version 26.0.0 (IBM Inc., Armonk, NY, USA) was used to carry out the confirmatory factor analysis. All procedures described in this study were approved by the Research and Ethics Committee of Nursing, Physiotherapy, and Medicine Department of the University of Almeria (Spain), with approval number EFM57b/2020. This study used secondary data sources for the literature review and validation studies; the questionnaire did not collect personal information, and respondents’ emails were not recorded.

## 3. Results

### 3.1. Evaluation by Expert Review

The item-level content validity indexes (I-CVIs) for relevance, likely effectiveness, and appropriateness for the intended audience were 1.00 for each core dimension of the pediatric website and the questionnaire. The proportion of items on the scale that achieved a relevance scale of 4 or 5 by all experts (S-CVI/AV) for the global content evaluation of both the website and the questionnaire was also 1.00. For the website, reviewers provided narrative comments about usability and the use of technical words. For the questionnaire, reviewers commented on the likely mechanisms of shortening questions, concerns about privacy aspects, and proposed minor changes in eight of the 41 qualitative items. The five participants who evaluated the feasibility, simplicity, and time required to answer the questionnaire stated that the questionnaire was easy to understand and easy to fill out in less than five minutes.

### 3.2. Demographic Characteristics of the Respondents

At the moment of performing this research, the website had been working for five years and six months and had been viewed 2,909,785 times by 2,432,167 unique visitors. Within the eight weeks that the questionnaire was available, the website was viewed 117,032 times by 98,577 unique visitors. As 516 responses were submitted, 0.5% of the visitors participated in this study. The mean age of the respondents was 38.8 years with a standard deviation (SD) of 6.1 years. The mean age of their youngest child was 4.6 years, with a SD of 4.0 years. The demographic characteristics of the respondents are shown in Table 1 and in Appendix A.

### 3.3. Construct Validity

Exploratory factor analysis. The KMO statistic was 0.783, and the result of Bartlett’s test was *p* < 0.001. The exploratory factor analysis excluded 27 of the 41 initial qualitative items from the questionnaire (Appendix A). The EFA of the resulting 14-item questionnaire identified five domains, which explained 74.7% of the variance in the data (Appendix A). Commonalities of the 14 items ranged from 0.630 to 0.840 (Appendix A). A five-factor model was chosen as the best analytic solution based on the factor loadings, parallel analysis, and conceptual knowledge (Table 2; Appendix A). Domain one was measured by three items, domain two by another three items, domain three by two items, domain four by four items, and domain five by two items. The model was validated to acknowledge the quality of the obtained solution comparing the initial correlation matrix with the matrix generated from the latent variables.

The resulting domains were interpreted by assigning them a name based on the original variables included in each domain. This confirmed that the five constructs underlying the 14-item instrument were usability (items 1–3), utility of the website (items 4–6), trust and confidence (items 7–8), utility of the well-child section (items 9–12), and accessibility (items 12–14). The global Cronbach’s alpha for the resulting questionnaire was 0.80. Cronbach’s alpha values were 0.64 for the trust and confidence construct, 0.74 for usability, 0.75 for accessibility, 0.81 for utility, and 0.89 for the well-child section. The split-half method did not detect significant differences in the domains or in the global evaluation. (Table 3).

### 3.4. Confirmatory Factor Analysis

The model showed 67 degrees of freedom, a Chi-square value of 290.43, and a probability level of *p* < 0.001 (Appendix A). Measures of internal consistency of the construct by confirmatory factor analysis are shown in Figure 1. The five constructs detected in EFC were confirmed, with the items presenting correlations from 0.62 to 0.93. Regression weights, standard errors, critical ratio, and significances were calculated (Appendix A). Critical ratio values were high, and the differences were significant in all the estimated parameters. The first latent variable, usability, was defined by the first three observational variables and their respective errors. All of them contribute similarly with regression values between 0.39 and 0.56. The second latent variable, utility of the website, was defined by the next three items, with regression values between 0.43 and 0.87. The third variable, trust and confidence, was defined by two items with regression values between 0.6 and 0.72. The fourth variable, the well-child section, was defined by four items with regression values between 0.43 and 0.64. The last latent variable, accessibility, was defined by two items with regression values between 0.46 and 0.81. The goodness of fit for the competing models was evaluated through fit indices.

A global evaluation of the model was performed to determine whether it properly replicated the existing relations in the data covariance matrix. The chi-square test evaluates the null hypothesis of a non-significant model. Our model showed 67 degrees of freedom, a chi-square value of 290.43, and a probability level of <0.001, a high significance value (Appendix A). The GFI value was 0.919. The value of the magnitude evaluation of the RMSEA was 0.08. The NFI was 0.907, the NNFI (or TLI) value was 0.900, and the CFI value was 0.926 (Appendix A).

## 4. Discussion

A website questionnaire was developed to evaluate different aspects of a pediatric website for parents from the point of view of users. The development of a valid and reliable assessment tool is not a trivial task [18]. Therefore, after studying other research in different medical ambits [16,17,19,23], an evaluation of the content validity and the psychometric reliability of the questionnaire was performed. The most important aspect of this research is that, as far as the authors know, this could be the first study to develop and test a website questionnaire to evaluate a pediatric website for parents.

### 4.1. Sample Size

Some authors propose a sample size between 50 and 100 individuals to carry out factor analysis and evaluate the psychometric properties of questionnaires [18], while other authors propose applying the classical rule of Kline [21] or using tools like Epi Info™ [20]. This tool indicates a required sample size of 471 participants, similar to the figures reported by other authors [21,24]. We collected 516 submitted questionnaires, well above the number calculated with the Epi Info™ tool and the number recommended by other authors [18,21].

### 4.2. Demographical Data

Respondents of our survey were mostly women, as it seems that women perform more health searches on the Internet [4,25,26,27]. Users who perform health searches on the website usually have a higher education level [26], something we have also found. It seems that users among 18–29 years old search more about health on the website [24], but the mean age of our respondents was higher, as our website is a parent-targeted site. People with more home income could tend to perform more health searches on the website [28], but there are also studies which find no differences [29]. Our results seem to be more in line with this last statement. Geographical disparities in access to primary care seem not to be associated with differences when using the Internet for health purposes [4,27], but authors state that people who live in urban areas performed more health searches in the Internet [8]. Among our respondents, most of them live in urban areas. Almost all of our respondents declared that they performed health-related searches on the website. Our figures are larger than the ones reported by other authors [30,31], but we should consider that our respondents accessed our questionnaire from a pediatric health web, so they probably are Internet active users, and they probably are concerned with the health of their children.

### 4.3. Content Validity

In our case, the I-CVI was a slightly more conservative indicator of expert consensus than described by other authors [16,19] because the survey tool utilized for the ratings of our questionnaire used a five-point scale rather than a four-point scale, with the aim of minimizing selection bias.

### 4.4. Construct Validation

The exploratory factor analysis allowed us to exclude 27 items from the initial 41-item questionnaire. The number of items excluded seems large, but it is in consonance with the research published by authors like Castro [32]. The exploratory factor analysis of our resulting 14-item questionnaire detected consistency in five grouped domains that were interpreted based on that which the items had in common, as stated by authors like Paiva in their research [18]. Similar to the work by other authors who evaluated questionnaires in other ambits [18,32,33], the factorial solution was able to explain the theoretical model.

### 4.5. Reliability

Cronbach’s alpha coefficient was used to evaluate the internal consistency of the questionnaire, which is the method employed in most validation studies found in the literature [18]. Although Cronbach’s alpha values were slightly under 0.7 for one construct (trust and confidence), for authors like Paiva and Sasaki, an Alpha value equal or higher than 0.67 is an almost acceptable value [33]. Values >0.80 are considered as good results for authors like Paiva [18] or excellent, according to Castro [32]. In our case, two of the five constructs were above 0.81, and the global Cronbach’s alpha for our final 14-item questionnaire was 0.80.

### 4.6. Confirmatory Factorial Analysis

As referred to by authors like Tsai, the confirmatory factorial analysis is deemed of methodological merits in terms of determining the underlying conceptual structure rigorously [34]. We proceeded to perform the confirmatory factor analysis by applying the maximum likelihood estimation method. The chi-square test has a high dependence on the sample size, but our sample consisted of 516 questionnaires, well above the figures recommended by other authors [18]. The magnitude evaluation of RMSEA is subjective, but values under 0.08 are considered indicative of good fit. In our case, this value was within the intended range. NFI values can range from 0 to 1, and those above 0.9 are considered acceptable, which was our case. TLI values can range between 0 and 1, and values above 0.9 are considered desirable, as was our case. CFI values can range between 0 for a bad model and 1 for a good model; in our case, the value was near to 1. Therefore, in our model, the CFI and the TLI are both within the desired minimum range. The NFI and GFI values are above the minimum desired value, and the RMSEA is within the desired range. These values allow us to affirm that, after the removal of 26 items from the initial 40-item questionnaire, the previously established domains were confirmed, and the evaluation by the maximum likelihood estimation method was good.

### 4.7. Limitations and Strengths

This research has some limitations. The main one is that our sample is a convenience sample. In open web-based surveys, selection bias occurs inevitably [17]. Within the eight weeks that the questionnaire was available, only 0.5% of the visitors of the website participated in our study. Therefore, we should not assume that our sample is representative of the users of our website. Our website and our questionnaire were developed in Spanish, and we cannot infer that our conclusions are applicable to other languages [22]. Content validity evaluation is a subjective procedure [35], and every expert could not represent all the dimensions of the content domain [19]. Our questionnaire has not been validated in other web pages of the same type, and this could reduce the external validity of the results obtained. The inclusion in the study sample of people of different nationalities could introduce some bias and limit the results, as transcultural adaptation of the questionnaire was not performed. As personal data were not collected, it was impossible to know whether the same person, or the same IP address, was able to send more than one questionnaire. Our research also has some strengths. The main one is that our sample consisted of 516 valid questionnaires, well above the figures recommended by other authors [18]. In addition, our questionnaire achieved good results in all the validation studies, and the results of the reliability tests were good.

## 5. Conclusions

The aim of the present study was to develop a questionnaire to evaluate users’ perceived utility of a Spanish pediatric website for parents and to acquire evidence of the dimensional structure and reliability of the questionnaire. After performing a factorial analysis, our findings offer evidence of the validity and reliability of a final 14-item questionnaire to evaluate different aspects (usability, utility, accessibility, a well-child section, and trust and confidence) of a pediatric website for parents in Spanish. A proper evaluation of eHealth interventions could potentially lead to an improvement of their effectiveness and reliability. Both are essential aspects of the development of any actual and future eHealth intervention that should be properly evaluated. Given the psychometric features of our questionnaire, we can assume that this instrument is a valid tool to evaluate a Spanish eHealth pediatric website for parents. Therefore, our questionnaire is a pilot test that can be helpful for the research of new questionnaires to evaluate eHealth interventions that are similar to ours.

## Figures and Tables

**Figure 1 ijerph-17-02671-f001:**
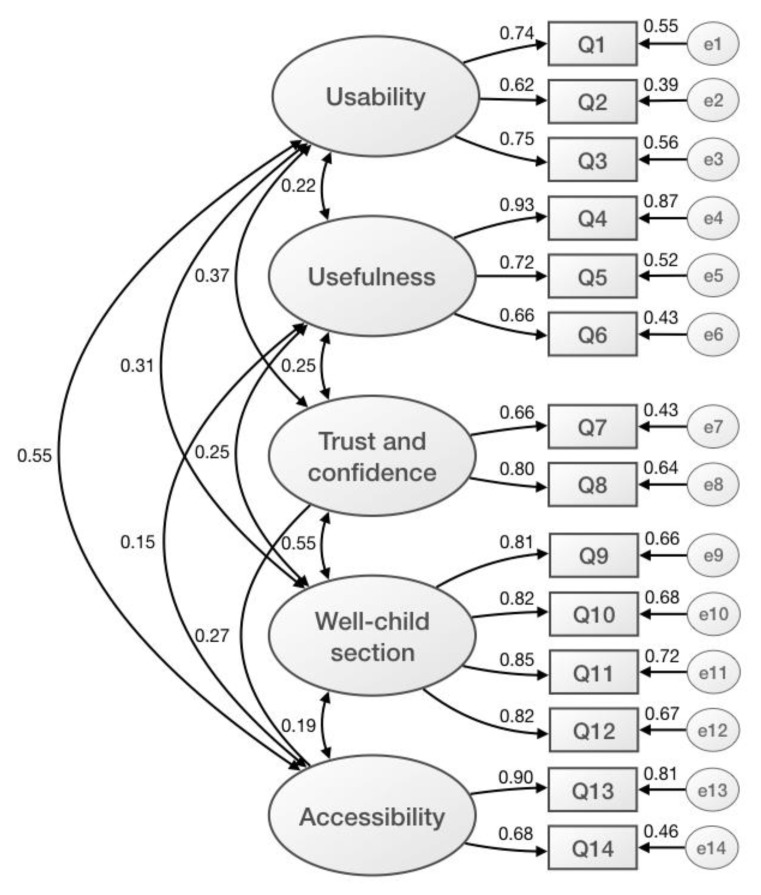
Measures of internal consistency of the construct by a confirmatory factor analysis (maximum likelihood estimation method).

**Table 1 ijerph-17-02671-t001:** Demographic characteristics of survey respondents.

Demographic Characteristics of Survey Respondents		
Respondent Characteristics	N	%
**Gender**		
Female	350	67.8
Male	166	32.2
**Number of children**		
No children	18	3.5
One	182	35.3
Two	277	53.7
Three or more	39	7.6
**Level of education**		
Primary / basic education	20	3.9
Secondary education / high school	92	17.8
University studies	253	49
Postgraduate / master’s	128	24.8
Others	23	4.5
**Home income**		
Less than €11,000 per year	44	8.5
Between €11,000 and €25,000 per year	133	25.8
Between €26,000 and €50,000 per year	190	36.8
Between €51,000 and €75,000 per year	92	17.8
More than €75,000 per year	57	11
**Geographical location**		
Spain	476	92.2
Central or South America	24	4.7
North America	12	2.3
Other European countries	3	0.6
Africa	1	0.2
**Geographical area**		
Urban	403	78.1
Rural	113	21.9
**Internet frequency access**		
Several times a day	492	95.3
Once a day	18	3.5
Each two or three days	6	1.2
Once a month or less	0	0
**Use of Internet for health searches**		
Yes	486	94.2
No	30	5.8

**Table 2 ijerph-17-02671-t002:** Factor rotation matrix.

Factor Rotation Matrix (a)	1	2	3	4	5
Item 1			0.757		
Item 2			0.775		
Item 3			0.825		
Item 4		0.880			
Item 5		0.846			
Item 6		0.800			
Item 7					0.852
Item 8					0.804
Item 9	0.894				
Item 10	0.883				
Item 11	0.825				
Item 12	0.786				
Item 13				0.829	
Item 14				0.896	

Extraction method: Principal factor analysis. Rotation method: Varimax rotation with Kaiser standardization. (a) Five-factor model.

**Table 3 ijerph-17-02671-t003:** Split-half method. Mean score comparison. Domains and global.

Split-Half Method. Mean Scores Comparison. Domains and Global.					
		N	Mean	SD	*p* Value
Domain 1	1st Half	258	12.76	1.75	0.77
	2nd Half	258	12.80	1.62	
Domain 2	1st Half	258	5.42	2.38	0.70
	2nd Half	258	5.51	2.45	
Domain 3	1st Half	258	5.70	0.74	0.82
	2nd Half	258	5.72	0.68	
Domain 4	1st Half	258	9.56	2.87	0.06
	2nd Half	258	9.90	2.42	
Domain 5	1st Half	258	9.38	1.02	0.86
	2nd Half	258	9.42	0.91	
Global	1st Half	258	42.52	5.80	0.35
	2nd Half	258	43.68	5.05	

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
