# Peer review of "Validation of a Questionnaire Developed to Evaluate a Pediatric eHealth Website for Parents"

_ijerph, 2020, doi:10.3390/ijerph17082671_

Round 1

Reviewer 1 Report

Thank you for the opportunity to review this paper on the validation of a questionnaire used to evaluate a paediatric eHealth website use by parents. The questionnaire was developed following the Cherries guidelines, and the Cherries checklist is provided as a supplementary file. The focus of the paper is on a detailed validation of the questionnaire using several methods and these appear to be appropriate.

The authors note that “that there is not a standard or universally accepted questionnaire to evaluate pediatric eHealth webs for parents”. This implies that there are existing questionnaires developed for this task. However, none are referred to in the paper, and the questionnaire developed by the authors is not compared to any other such assessment tool.

The authors note that this is the only such tool available in Spanish. Is this the novelty of the study?

Review Board Approval:

  • is it correct to say that the study used secondary data sources? Surely, the authors gathered and analysed primary data using their questionnaire and Google forms?
  • were the respondents’ email automatically addresses recorded within Google forms?

Unless it is the Journal’s style, probability should not be reported as 0.000. P=0.000 suggests that there is absolutely no chance of getting the results if the null hypothesis is true. P=0.000 is usually the result of software rounding up numbers. It is generally advised to reported it as p<0.001.

Discussion, demographical data: the word ‘preferably’ appears to have been used incorrectly in the sentence, “Respondents in our survey were preferably women..”. It appears that what is meant is that they were ‘mostly’ women.

Conclusions: line 247: the sentence refers to a 14-item questionnaire when, in fact, it was a 41-item questionnaire with an additional 26 demographical questions.

Minor queries:

When data are collected as whole numbers, for example, respondents’ age, their average should be presented only to one decimal place.

There are several minor spelling, typographical and grammatical errors.

Author Response

Dear reviewer,

Thank you for your kind and exhaustive review of the article. We are aware that our research has some important limitations, so we have thoroughly read your suggestions and have corrected all the subjects that were adressed.

We include an explanation of the modifications performed, in this letter.

We hope that our responses and modifications have improved the quality of the article, and that they satisfy your requests.

Thank you very much for your valuable time.

————————————————————

Reviewer 1.

Thank you for the opportunity to review this paper on the validation of a questionnaire used to evaluate a paediatric eHealth website use by parents. The questionnaire was developed following the Cherries guidelines, and the Cherries checklist is provided as a supplementary file. The focus of the paper is on a detailed validation of the questionnaire using several methods and these appear to be appropriate.

The authors note that “that there is not a standard or universally accepted questionnaire to evaluate pediatric eHealth webs for parents”. This implies that there are existing questionnaires developed for this task. However, none are referred to in the paper, and the questionnaire developed by the authors is not compared to any other such assessment tool.

Answer:

We wrote that sentence in a way that could create confusion. The proper sentence is «we have not found validated questionnaires designed to evaluate pediatric web for parents». We have modified the text (lines 45-46).

The authors note that this is the only such tool available in Spanish. Is this the novelty of the study?

Answer:

As far as we know, yes, but it is right that we have not properly reflected it. We have rewritten this sentence in Discussion (line 196): «The main novelty of this research is that, as far as the authors know, this is the first study to develop and test a web questionnaire to evaluate a Spanish pediatric web for parents».

Review Board Approval:

Is it correct to say that the study used secondary data sources? Surely, the authors gathered and analysed primary data using their questionnaire and Google forms?

Answer:

The secondary data sources were used for literature review and validation studies, and the questionnaire did not collect personal information. As the original sentence could create confusion, we have modified it, as follows (lines 113-116): «This study used secondary data sources for literature review and validation studies, and the questionnaire did not collect personal information and respondents’ email were not recorded. Although Research Ethical Committee and Institutional Review Board approval was not necessary, we requested it».

were the respondents’ email automatically addresses recorded within Google forms?

Answer:

As respondents’ email were not recorded within Google forms, we have added that sentence (lines 114-115): «Respondents’ email were not recorded».

Unless it is the Journal’s style, probability should not be reported as 0.000. P=0.000 suggests that there is absolutely no chance of getting the results if the null hypothesis is true. P=0.000 is usually the result of software rounding up numbers. It is generally advised to reported it as p<0.001.

Answer:

It was a mistake, we have corrected it (lines 144 and 168).

Discussion, demographical data: the word ‘preferably’ appears to have been used incorrectly in the sentence, “Respondents in our survey were preferably women..”. It appears that what is meant is that they were ‘mostly’ women.

Answer:

We have corrected that, as suggested (line 204).

Conclusions: line 247: the sentence refers to a 14-item questionnaire when, in fact, it was a 41-item questionnaire with an additional 26 demographical questions.

Answer:

In that line we refer to the final questionnaire, that included 14-items, after performing exploratory factorial analysis that allowed us to exclude 27 of the initial 41-qualitative items. As the sentence could create confusion, we hace changed it, as follows (lines 273-274): «After performing factorial analysis, our findings offer evidence of the validity and reliability of a final 14-item questionnaire».

Minor queries:

When data are collected as whole numbers, for example, respondents’ age, their average should be presented only to one decimal place.

Answer:

We have changed that, as requested (lines 131-133).

There are several minor spelling, typographical and grammatical errors.

Answer:

We have revised the manuscript to correct spelling, typographical and grammatical errors, as requested.

Reviewer 2 Report

  • line 44:  "...for parents. he" by "...for parents. The"
  • line 57: " they were were..." by "they were"

Experimental section.

When authors say: "The questionnaire was made accessible through a link posted in twenty-five pages of the pediatric web.", is necessary an explanation about the mechanism to obtain information in relation with replies to their questionnaire. And too and explanation of how authors filter this information.

I am concerned about the possibility that the same person, or the same IP address, have been able to reply to the questionnaire.

Exist in this situation or not? and in case that exists how many questionnaires were discarded? 

Ethical concern

It was impossible for me to find the ethical section in this article. Don't exist reference about the review and approval of this study by no one Research Ethical Committee (REC) or the existence of an explicit participant acceptation to participate in the study.

With the information offered in their article, I can suppose only that authors applied the questionnaire and recollect personal information without having the REC's approbation and this is considered as a big flaw that invalidates their study.

Author Response

IJERPH Review 20200327. Reviewer 2.

Dear reviewer,

Thank you for your kind and exhaustive review of the article. We are aware that our research has some important limitations, so we have thoroughly read your suggestions and have corrected all the subjects that were adressed.

We include an explanation of the modifications performed, in this letter.

We hope that our responses and modifications have improved the quality of the article, and that they satisfy your requests.

Thank you very much for your valuable time.

————————————————————

Reviewer 2.

line 44:  "...for parents. he" by "...for parents. The"

Answer:

Lines 45-46. Typographical mistake, corrected as requested.

line 57: " they were were..." by "they were"

Answer:

Line 59. Typographical mistake, corrected as requested.

Experimental section.

When authors say: "The questionnaire was made accessible through a link posted in twenty-five pages of the pediatric web.", is necessary an explanation about the mechanism to obtain information in relation with replies to their questionnaire. And too and explanation of how authors filter this information.

Answer:

Personal data was not collected by us or by Google forms. The answers of the questionnaire were included in a spreadsheet generated by Google forms, and we used that data to perform the analyses. We did not need to modify that data or to perform statical corrections, as only fully completed questionnaires were admitted.

To explain this, we have added the following text (lines 75-79): «Only fully completed questionnaires were admitted. Personal data was not collected within the questionnaire or Google forms platform. Google forms generated a spreadsheet with the answers of the questionnaires. The data collected within that spreadsheet was used, without any manipulation or statistical correction, to perform the statistical analyses. No questionnaires needed to be discarded».

I am concerned about the possibility that the same person, or the same IP address, have been able to reply to the questionnaire.

Answer:

Indeed, as personal data was not collected, it was impossible to know if the same person, or the same IP address, was able to send more than one questionnaire. We have added this sentence in limitations (lines 263-265): «as personal data was not collected, it was impossible to know if the same person, or the same IP address, was able to send more than one questionnaire».

Exist in this situation or not? and in case that exists how many questionnaires were discarded? 

Answer:

Only fully completed questionnaires could be submitted, so we did not have to discard any incomplete questionnaires. In Methods, we have added the sentence (line 79): «No questionnaires needed to be discarded».

Although we think it is not probable, we were not able to know if the same person submitted more than one questionnaire (text added in lines 263-265: «as personal data was not collected, it was impossible to know if the same person, or the same IP address, was able to send more than one questionnaire»).

Ethical concern

It was impossible for me to find the ethical section in this article. Don't exist reference about the review and approval of this study by no one Research Ethical Committee (REC) or the existence of an explicit participant acceptation to participate in the study.

With the information offered in their article, I can suppose only that authors applied the questionnaire and recollect personal information without having the REC's approbation and this is considered as a big flaw that invalidates their study.

Answer:

The secondary data sources were used for literature review and validation studies, and the questionnaire did not collect personal information. As the original sentence included in the article about this important subject could create confusion, we have modified it, as follows (lines 113-116): «This study used secondary data sources for literature review and validation studies, and the questionnaire did not collect personal information and respondents’ email were not recorded. Although Research Ethical Committee and Institutional Review Board approval was not necessary, we requested it».

Finally, we want to state that an informed consent, showing all this important information about the study and the kind of data collected, was displayed at the initial page of the questionnaire, so this sentence has been added in lines 75-76: «Informed consent was shown at the initial page of the questionnaire.» 

Round 2

Reviewer 1 Report

Thank you for the opportunity to re-review this paper. Most of the queries have been satisfactorily addressed. The responses lead to another query.

Previous query:

The authors note that “that there is not a standard or universally accepted questionnaire to evaluate pediatric eHealth webs for parents”. This implies that there are existing questionnaires developed for this task. However, none are referred to in the paper, and the questionnaire developed by the authors is not compared to any other such assessment tool.

Response

We wrote that sentence in a way that could create confusion. The proper sentence is «we have not found validated questionnaires designed to evaluate pediatric web for parents». We have modified the text (lines 45-46).

The authors note that this is the only such tool available in Spanish. Is this the novelty of the study?

Response

As far as we know, yes, but it is right that we have not properly reflected it. We have rewritten this sentence in Discussion (line 196): «The main novelty of this research is that, as far as the authors know, this is the first study to develop and test a web questionnaire to evaluate a Spanish pediatric web for parents».

New query:

While the second response emphasises the Spanish component of the novelty, it does not take into account the first response, which implies that this is the first study to validate a tool for evaluation of a paediatric website for parents. If this is correct, this is then the most important aspect of the study.

Review Board Approval:

Previous query:

Is it correct to say that the study used secondary data sources? Surely, the authors gathered and analysed primary data using their questionnaire and Google forms?

Response The secondary data sources were used for literature review and validation studies, and the questionnaire did not collect personal information. As the original sentence could create confusion, we have modified it, as follows (lines 113-116): «This study used secondary data sources for literature review and validation studies, and the questionnaire did not collect personal information and respondents’ email were not recorded. Although Research Ethical Committee and Institutional Review Board approval was not necessary, we requested it».

New Query:

If approval was requested, as stated in the response, was it granted? If it was, then this should be stated in the paper, preferably with the approval number. It remains unclear as to why, in the first version, you said it was not necessary, even though you had applied for it

Minor queries:

When data are collected as whole numbers, for example, respondents’ age, their average should be presented only to one decimal place.

We have changed that, as requested (lines 131-133).

New query: This has not been done correctly in all instances.

Line 126: 0.52 should be 0.5.

Line 127: 6.06 rounded up, should be 6.1 and not 6.0.

Line 136: should 74.68, not be rounded to 74.7%.

Spelling in the paper still needs to be carefully checked, for example:

Line 121: “less tan five minutes”.

Author Response

Dear reviewer,

Again, thank you for your kind review of the article. We have thoroughly read your suggestions and have corrected all the subjects that were adressed.

We include an explanation of the modifications performed, in this letter.

We hope that our responses and modifications have improved the quality of the article, and that they satisfy your requests.

Thank you very much for your valuable time.

————————————————————

Reviewer 1.

Thank you for the opportunity to re-review this paper. Most of the queries have been satisfactorily addressed. The responses lead to another query.

Previous query:

The authors note that “that there is not a standard or universally accepted questionnaire to evaluate pediatric eHealth webs for parents”. This implies that there are existing questionnaires developed for this task. However, none are referred to in the paper, and the questionnaire developed by the authors is not compared to any other such assessment tool.

Response

We wrote that sentence in a way that could create confusion. The proper sentence is «we have not found validated questionnaires designed to evaluate pediatric web for parents». We have modified the text (lines 45-46).

The authors note that this is the only such tool available in Spanish. Is this the novelty of the study?

Response

As far as we know, yes, but it is right that we have not properly reflected it. We have rewritten this sentence in Discussion (line 196): «The main novelty of this research is that, as far as the authors know, this is the first study to develop and test a web questionnaire to evaluate a Spanish pediatric web for parents».

New query:

While the second response emphasises the Spanish component of the novelty, it does not take into account the first response, which implies that this is the first study to validate a tool for evaluation of a paediatric website for parents. If this is correct, this is then the most important aspect of the study.

We have modified the text to reflect this more precisely (lines 200-201): «The most important aspect of this research is that, as far as the authors know, this could be the first study to develop and test a web questionnaire to evaluate a pediatric web for parents».

Review Board Approval:

Previous query:

Is it correct to say that the study used secondary data sources? Surely, the authors gathered and analysed primary data using their questionnaire and Google forms?

Response

The secondary data sources were used for literature review and validation studies, and the questionnaire did not collect personal information. As the original sentence could create confusion, we have modified it, as follows (lines 113-116): «This study used secondary data sources for literature review and validation studies, and the questionnaire did not collect personal information and respondents’ email were not recorded. Although Research Ethical Committee and Institutional Review Board approval was not necessary, we requested it».

New Query:

If approval was requested, as stated in the response, was it granted? If it was, then this should be stated in the paper, preferably with the approval number. It remains unclear as to why, in the first version, you said it was not necessary, even though you had applied for it.

We are sorry that we did not state this aspect clearly, previously. Despite Review Board Approval was not necessary in our research, we were aware that it was recommended, so we requested evaluation and it was granted, as all procedures described in this research were approved. Indeed, we understand that we had not stated it clearly, so we have modified the text to reflect it, as suggested (lines 111-117): «All procedures described in this study were approved by the Research and Ethics Committee of Nursing, Physiotherapy and Medicine Department of University of Almeria (Spain), with approval number EFM57b/2020. This study used secondary data sources for literature review and validation studies, and the questionnaire did not collect personal information and respondents’ email were not recorded».

Minor queries:

When data are collected as whole numbers, for example, respondents’ age, their average should be presented only to one decimal place.

We have changed that, as requested (lines 131-133).

New query: This has not been done correctly in all instances.

Line 126: 0.52 should be 0.5.

Line 127: 6.06 rounded up, should be 6.1 and not 6.0.

Line 136: should 74.68, not be rounded to 74.7%.

We have modified these, as requested (lines 131-132, and 147).

Spelling in the paper still needs to be carefully checked, for example:

Line 121: “less tan five minutes”.

We have modified that, as requested (line 126).

Reviewer 2 Report

Dear Authors

The introduction of changes in your manuscript expresses clearly the situation related to your research project ethical concerns.

Thank you.

Author Response

Dear reviewer,

Once more, thank you for your kind review of the article. We have thoroughly read your suggestions and have corrected all the subjects that were adressed.

We include an explanation of the modifications performed, in this letter.

We hope that our responses and modifications have improved the quality of the article, and that they satisfy your requests.

Thank you very much for your valuable time.

————————————————————

Reviewer 2.

Dear Authors

The introduction of changes in your manuscript expresses clearly the situation related to your research project ethical concerns.

Thank you.

We are sorry that we did not state clearly that approval of Research and Ethics Committee evaluation was requested and granted. Indeed, we have modified the text to clearly reflect this: «All procedures described in this study were approved by the Research and Ethics Committee of Nursing, Physiotherapy and Medicine Department of University of Almeria (Spain), with approval number EFM57b/2020. This study used secondary data sources for literature review and validation studies, the questionnaire did not collect personal information and respondents’ email were not recorded».

This manuscript is a resubmission of an earlier submission. The following is a list of the peer review reports and author responses from that submission.